# Comparison between Full Body Motion Recognition Camera Interaction and Hand Controllers Interaction used in Virtual Reality Exposure Therapy for Acrophobia

**DOI:** 10.3390/s20051244

**Published:** 2020-02-25

**Authors:** Jacob Kritikos, Chara Zoitaki, Giannis Tzannetos, Anxhelino Mehmeti, Marilina Douloudi, George Nikolaou, Giorgos Alevizopoulos, Dimitris Koutsouris

**Affiliations:** 1School of Electrical and Computer Engineering, National Technical University of Athens, 15780 Athens, Greece; charazoitaki@gmail.com (C.Z.); tzannetosg@gmail.com (G.T.); 2Department of Informatics and Telecommunications, National Kapodistrian University of Athens, 15784 Athens, Greece; jinomehmeti@gmail.com (A.M.); gs.nikolaoy@gmail.com (G.N.); 3Department of Biology, National Kapodistrian University of Athens, 15784 Athens, Greece; marilina.douloudi@gmail.com; 4Agioi Anargyroi, General & Oncological Hospital, 14564 Athens, Greece; galev@nurs.uoa.gr

**Keywords:** virtual reality, cognitive behavioral therapy, exposure therapy, anxiety disorders, specific phobias, acrophobia, motion tracking sensor, motion recognition camera, hand controllers, presence, immersion

## Abstract

Virtual Reality has already been proven as a useful supplementary treatment tool for anxiety disorders. However, no specific technological importance has been given so far on how to apply Virtual Reality with a way that properly stimulates the phobic stimulus and provide the necessary means for lifelike experience. Thanks to technological advancements, there is now a variety of hardware that can help enhance stronger emotions generated by Virtual Reality systems. This study aims to evaluate the feeling of presence during different hardware setups of Virtual Reality Exposure Therapy, and, particularly how the user’s interaction with those setups can affects their sense of presence during the virtual simulation. An acrophobic virtual scenario is used as a case study by 20 phobic individuals and the Witmer–Singer presence questionnaire was used for presence evaluation by the users of the system. Statistical analysis on their answers revealed that the proposed full body Motion Recognition Cameras system generates a better feeling of presence compared to the Hand Controllers system. This is thanks to the Motion Recognition Cameras, which track and allow display of the user’s entire body within the virtual environment. Thus, the users are enabled to interact and confront the anxiety-provoking stimulus as in real world. Further studies are recommended, in which the proposed system could be used in Virtual Reality Exposure Therapy trials with acrophobic patients and other anxiety disorders as well, since the proposed system can provide natural interaction in various simulated environments.

## 1. Introduction

Anxiety disorders are common mental health disorders that 19.1% of adults and 31.9% of adolescents in the US suffer from, with 22.8% of them reported as serious cases [1]. As a result of its physical manifestations, anxiety disorders cost an estimated $42 billion per year in the US, approximately one third of the entire US health bill [2]. There are many different types of anxiety disorders but they all share features of excessive fear, as the emotional response to real or perceived imminent threat, and anxiety, for anticipation of future threat. As a result, the suffering individual engages in abnormal behavior, with surges for fight, thoughts of immediate danger, and escape behaviors, as well as muscle tension and vigilance in preparation for future danger. Sometimes the level of fear or anxiety is reduced by pervasive avoidance behaviors, such as panic attacks [3]. Such behavior can hinder the individual from leading a normal everyday life, affecting their personal and professional interactions. Acrophobia is an extreme fear of heights and is considered a specific phobia of naturalistic type. It appears closely related to the fear of elevators, stairs, and fear of flying, both of which belong to the specific phobia category [3,4]. Individuals suffering from acrophobia typically avoid height-related situations such as stairs, terraces, apartments, and offices located in high buildings, bridges, elevators, and even plane trips [5]. Studies suggest that one in 20 people reach diagnostic criteria for acrophobia [6,7]. 

There are many different techniques of treatment for Anxiety Disorders but one of the most widely-used is Exposure Therapy (ET), which is a cognitive behavioral technique [8]. Exposure Therapy aims not only to alter the patient’s behavior but also the way they perceive their phobia [9]. Patients repeatedly exposed to the avoided stimuli (thoughts, objects, or situations) in a therapeutic manner, have shown a decline in anxiety symptoms over time, putting their beliefs surrounding the feared stimuli to test. Finally, they show improved self-efficacy in the face of perceived threat [10], [11]. Even though ET is widely used, there are some technical difficulties regarding its application: it can be costly for the clinician, in order to acquire the feared stimuli (animals, items, or places where the stimuli is most prominent), and time-consuming to apply, as well as costly for the patient. Therefore, in order for the better treatments to be used, they need to be enhanced in a way that will benefit both the patient and the clinician [12,13,14]. 

Over the past years, Virtual Reality (VR) technology has developed significantly. With VR products being more accessible, from a financial point of view [15,16] it became possible for more specific VR applications to be created. So, research has been done in order to determine whether Virtual Reality can improve Standard ET [16]. This is how Virtual Reality Exposure Therapy (VRET) was created. In VRET, patients are exposed to virtual, lifelike anxiety-provoking environments instead of real anxious situations [17,18]. By using VRET, it is possible to simulate phobic environments of any sort with equal ease, diminishing the aforementioned technical difficulties posed by Standard ET [19]. So, thanks to VR technologies, clinicians can deliver to their patients the most powerful element of direct therapeutic intervention—that is direct coaching in everyday situations that trouble them, an element frequently missing from clinics. By putting on a headset, patients can be taken immediately into various situations, graded in difficulty, where they are exposed to the cause of their psychological distress. Thanks to the safety provided by VR simulated environments, patients feel more comfortable and confident, and they are willing to face situations that trouble them and try alternative ways of responding. And, even though their actions take place in the virtual world, Freeman et al. argue that the learning transfers to the real world [12]. Therefore, treatment delivered using VR consumer hardware could become a low-cost way of providing effective interventions at scale. Over the last few years, studies have been conducted on the effects of VRET, and it was found to be as effective as Standard ET [20]. 

## 2. Related Work

Since the 1990s researchers have conducted multiple studies with acrophobic individuals using VRET, and, as a result, nowadays, there is considerable implication that VRET could be an effective means of acrophobia treatment [21,22,23,24,25,26]. However, the technical specifications of the systems used in the aforementioned studies can be considered primitive in comparison to modern ones regarding vividness, realism, emotion stimulation. The capabilities of current technology can help develop even more immersive virtual environments and simulations for potential use in a variety of VRET treatments. During some VRET simulations, the phobic individual simply watches a recorded fearful situation taking place in the virtual environment, without being able to participate actively, even in recent studies [27,28,29,30]. In simulations without interaction they can only identify the phobic situation but not directly confront and face the phobic stimuli from an egocentric or an allocentric point of view. In some VRET simulations basic interaction is provided: user input could be strictly gaze-derived [31], while motion feedback is provided via standard interfaces (i.e., mouse, keyboard, joystick [32], gamepad [8]). This way, the user at least controls the events of the simulation and does not remain a passive viewer of anxiety-inducing situations. However, recently it has been underlined that modern VR systems used in VRET should involve the user actively during the simulations by leveraging on motion tracking technologies [33,34,35]. Companies that produce VR device packages, such as Oculus and HTC, use trackable hand controllers to provide in-system interaction and presence to the user. So, by holding the controllers while the user moves their hands, they can see their virtual hands move in the virtual environment in real time [36,37]. There have even been efforts to include wearable motion tracking in VR systems, such as a wearable glove in [38]. Based on theories, any improvement in VRET depends on its ability to adequately initiate fear responses in fearful participants [8]. Key factors of the ability of VR to stimulate fear are the concepts of presence and immersion [17]. On the one hand, presence is a psychological state or subjective perception in which even though part or all of an individual’s current experience is generated by and/or filtered through human-made technology, part or all of the individual’s perception fails to accurately acknowledge the role of the technology in the experience [39], thus it is a subjective response to a given virtual environment. Immersion, on the other hand, is “a psychological state characterized by perceiving oneself to be enveloped by, included in, and interacting with an environment that provides a continuous stream of stimuli and experiences” [40]. In general, it is assumed that an increase in immersion -and therefore the improvement of the included technologies- increases the experienced presence, and that an increase in presence leads to stronger fear responses during VRET. Lately, various implementations of VR technology have shown promising improvement regarding immersion and therefore presence of the users in the system [41]. 

## 3. Research Question

In order to improve VRET, we present and propose a more immersive VR system, by adding a motion recognition system that consists of a body recognition camera, a hand recognition camera and a VR headset. By using motion recognition sensors, we manage to avoid interfaces that require unnatural confrontation (i.e., hand controllers, joysticks, etc.) in order to simulate movement and actions, thus allowing the user to feel that they are moving and interacting with their whole body within the virtual environment, in real time, as they would in the real world. In more technologically oriented studies, the use of various tracking tools that can help improve the human experience in the virtual augmented or mixed reality world have been thoroughly studied [42,43,44,45,46]. Nevertheless, there is still much room for research on how to apply these technologies in the fields related to mental disorders and, more specifically, what tool can be used in diverse types of treatment. 

In our original work we used only a body recognition camera [47], a demonstration video is available here: youtube.com/watch?v=yEeUvd1-2qQ. In our current work we use body and hand recognition cameras, a demonstration video is available here: youtube.com/watch?v=a0tjB6PMNjg. Therefore, we allow the user to communicate with the system through a most natural user interface, turning their physical actions into data that serve the motion tracking purposes, without adding more wearable equipment that would complicate their movement. 

In this study, we compare the aforementioned proposed system with commercially available VR systems which include hand controllers to provide interaction with the virtual environment—in terms of the feeling of presence the user experiences. Our goal is to conclude whether the proposed system enhances immersion and presence and therefore could be of use in future VRET that will involve patients diagnosed with an anxiety disorder, to determine whether it can improve related treatments. In our study, we use a case of acrophobia treatment.

## 4. System Description (Hardware and Software)

In both systems, we used the same Desktop Computer as basic equipment that provides the infrastructure to run both systems. Its specifications are: Graphics Card: NVIDIA GeForce GTX 1070, CPU: AMD Ryzen 7 2700X, RAM: 16 GB G.Skill TridentZ DDR4, Video Output: HDMI 1.3, USB Ports: 3 × USB 3.0 and 1 × USB 2.0. In addition, in both systems, an Oculus Rift Virtual Reality Headset provides the virtual environment and two Oculus Rift Sensors track constellations of IR LEDs to translate the movement of the headset within the virtual environment. Peripheral Hardware of the Virtual Reality-Hand Controllers (VR-HC) system: (Figure 1) The system includes two Oculus Rift Touch Controllers, one for each hand, which provide intuitive hand presence and interaction within the virtual environment. Again, the Oculus Rift Sensors track constellations of IR LEDs to translate the movement of the hand controllers within the virtual environment. 

Peripheral Hardware of the Virtual Reality-Motion Camera (VR-MC) system: (Figure 2) In this system, we use the Astra S model designed by Orbbec, which is a Body Motion Recognition Camera (BMRC). The BMRC is connected to the computer with a USB 2.0 cable. The motion tracking accuracy has an error range of ±1–3 mm from a 1 m distance, whereas, at a 3 m distance, it is estimated at approximately ±12.7 mm. The optimized maximum range is about 6m. In detail, Astra S consists of 2 cameras: a Depth Camera, with image size 640 × 480 (VGA) @ 30FPS and an RGB Camera, with image size 1280 × 720 @ 30FPS (UVC Support), which have 60° horiz × 49.5° vert. (73° diagonal) field of view. As a result of its tracking range, the BMRC is not designed to track movements of small body parts, such as the fingers. For that purpose, we also used the Leap Motion 3D Controller designed by Leap Motion Inc, which is a Hand Motion Recognition Camera (HMRC). The HMRC has 135° field of view and up to 120 fps and 8 cubic feet interactive 3D space. The HMRC is attached to the front VR headset.

The same software tools are used in both systems. Specifically: (a) Operating System: Windows 10, with the drivers for the Astra S, Leap Motion, and Oculus Rift installed; (b) Unity 3D, which is used as the basic program for creating the virtual environment. All the hardware pieces (i.e., sensors, controllers, trackers, headset, camera, etc.) feed Unity 3D with data, which affect the virtual environment that will be presented through the VR headset; (c) Blender 3D Computer Graphics Software is used for creating 3D objects, animated visual effects, UV Mapping, and materials integrated in Unity 3D; (d) Adobe Photoshop is used for creating images for the materials which are inserted in the Blender program; (e) OVRPlugin is used for the operation of the Oculus Rift equipment in Unity 3D; (f) Leap Motion Core Assets is used for the operation of the Leap Motion Unity 3D; (g) Nuitrack SDK as the motion recognition middleware between the developed software and the BMRC. Although we use Unity 3D in order to recreate simulations in both systems, for each system there is a different source file, because of the different way the user interacts within each system. Otherwise, both virtual rooms look the same and include the same objects (Figure 3). We want to keep the minimum amount of differences between both systems, because we aim to compare the two different interaction technologies, so, the rest of the system, both from the hardware and the software point of view, has to be identical. In this study, the user experiences and sees the same virtual room in both systems, apart from the way they interact within each environment. 

## 5. Body Recognition

In order to track the user’s position during the simulations, the user’s body is mapped onto a virtual skeleton that consists of spheres and lines: the spheres represent the joints and the lines represent the bones between joints. It is important to note that the anatomy of the virtual skeleton is not identical to that of the human body, but it serves its purpose to track the user’s movements. Each joint is essentially a point in 3D space represented by 3 coordinates: x, y, and z. The *y*-axis is the height axis and the x-z plane is the floor where the user stands and moves (Figure 4). The BMRC is placed across the user at the beginning of the axes. Therefore, the user can walk on the x-z plane, and, as long as they remain within the tracking field of the BMRC, and face it, they can see their virtual body appear and move in accordance to their physical movements in the virtual environment. The Astra S BMRC is designed for skeleton tracking and recognizes the following 17 joints of the human body: HEAD, NECK, TORSO, LEFT_COLLAR, LEFT_SHOULDER, RIGHT_SHOULDER, WAIST, LEFT_HIP, RIGHT_HIP, LEFT_ELBOW, RIGHT_ELBOW, LEFT_WRIST, RIGHT_WRIST, LEFT_KNEE, RIGHT_KNEE, LEFT_ANKLE, RIGHT_ANKLE (Figure 5). By design, it cannot track movements of smaller parts of the body, such as fingers, or the rotation of the hands.

Considering that the realism of the user’s experience is considered to affect their feeling of presence, in order to present a more complete representation of the virtual skeleton of the user as they move within the virtual environment, it was critical to include the Leap Motion HMRC to the system, which can recognize the following 25 joints of the human hand and wrist: (a) 2 joints for the wrist—one for each edge; (b) 1 joint for the palm; (c) 4 joints to mark the edges of the distal, intermediate, and proximal phalanges of each finger; (d) 2 joints to mark the edge of the metacarpal phalanges (Figure 6). 

Both the Body Motion Recognition Camera (BMRC) and the Hand Motion Recognition Camera (HMRC) track the position of the wrist. In order to avoid any conflicts regarding the representation of the position of the wrists, and, therefore, the rest of the hand, in the virtual environment, we have set a higher priority for the position tracked by the Leap Motion HMRC. That means that if both the HMRC and the BMRC receive information about the position of the wrist, the system uses the one received by the HMRC. The position information of the BMRC is used only if the system receives no data from the HMRC, which can occur in case the user’s hands are not in the field of the HMRC. 

## 6. System Setup 

The outlined green area is about 5m^2^ and will be referred to as the “VR-HC User Action Area”. Within this area, the user wears the Oculus Rift Virtual Reality Headset and holds the Oculus Rift Touch Controllers, one in each hand, with which they interact within the virtual environment. In front of the VR-HC User Action Area, two Oculus Rift Sensors have been placed, approximately 3 meters apart from each other and rotated appropriately in order to face the VR-HC User Action Area. Last, the Desktop Computer is placed on a table in front of the VR-HC User Action Area, and all cables from the peripheral devices end up there, for the operation of the system (Figure 7). The outlined area is about 5 m^2^ and will be referred to as the “VR-MC User Action Area”. Within this area, the user wears the Oculus Rift Virtual Reality Headset, on which we have attached the HMRC (Figure 7). The user, unlike in the VR-HC system, does not need to hold controllers or have any piece of hardware attached to their body in order to interact within the virtual environment, apart from the VR headset. The user can move and interact freely within the virtual room, while the BMRC and the HMRC track their movements. The Motion Recognition Camera is placed 1 m in front of the VR-MC User Action Area. Last, the Desktop Computer is placed on a table in front of the VR-MC User Action Area, and all cables from the peripheral devices end up there, for the operation of the system (Figure 8).

## 7. Method and Procedure 

We investigated the differences in presence between the VR-HC and VR-MC systems in conjunction with two experiments, one for each system, in a 45-minute session. A total of 20 people (12 men and 8 women, ages 18–30, avg. age 24) served as participants. They were recruited in the National Technical University of Athens campus in Zografou, Athens, Greece and diagnosed with acrophobia symptoms by Psychiatrists, Professor George Alevizopoulos MD. The study was approved by the Ethics Committee of the National Technical University with protocol number #51105. 

All 20 participants used both the systems, successively. In order to avoid the possibility that the order of using the systems could make any difference, the procedure was randomized: half of the participants first used the VR-HC and then the VR-MC system, and the other half in the reverse order. This assignment was performed randomly upon recruiting each participant. Both experiments required participants to perform the same simple tasks of traversing a room, going out on a balcony and touching an object that was hanging from the ceiling at a considerable distance from the balcony, by using the moving mechanisms of each of the two systems, i.e., the Hand Controllers and the Motion Recognition System. The purpose of the study is to compare how different interaction methods affect the feeling of presence of the user. Therefore, the virtual environment was designed simply and in the same way in both systems, since it is not the issue under study and we did not want the participants to focus on it.

The study took place at the Biomedical Engineering Laboratory of the National Technical University of Athens, Greece. All of the 20 participants performed the same procedure, during a single 45-minute session. The aim of the session was to use both the VR-HC and VR-MC systems and then compare their experiences by answering a Witmer–Signer presence questionnaire for each system. The session consisted of 5 phases: Phase 1: The participant was informed about the study and its procedure (10 mins).Phase 2: The simulation of the first system took place (5 mins).Phase 3: A short break so that the participant takes time to process their experience (3 mins).Phase 4: The participant filled out the respective Presence Questionnaire (10 m ins).Phase 5: The simulation of the second system took place (5 mins).Phase 6: A short break so that the participant takes time to process their experience (3 mins).Phase 7: The participant filled out the respective Presence Questionnaire (10 mins).

The participant, in this simulation, wears the Oculus Rift Virtual Reality Headset, holds the Hand Controllers and goes to the center of the VR-HC User Action Area, where they stand until the simulation starts (Figure 9). Then, the participant will find themselves in a virtual environment; an apartment room, located on a high floor, with an open balcony door. The participant can see the controls that they are holding in the real world materialize in the virtual room and move in real-time, according to the actual movements of their hands. With them, they can interact within the virtual room. The participant has three consecutive tasks to execute. They have to walk across the room, towards the open balcony door and get out, on the balcony, touch the railing of the balcony with their two controllers and stay in this state for a few minutes. Then, they have to catch the object hanging before them, at a significant distance ahead of the balcony railing. When they do so, they can turn back, inside the room, and the task is considered completed. Last, the participant removes the Headset and the Controllers and returns to the real world.

The participant, in this simulation, wears the Oculus Rift Virtual Reality Headset and goes to the center of the VR-MC User Action Area, where they stand until the simulation starts (Figure 10). Then, the participant will find themselves in a virtual environment; an apartment room, located on a high floor, with an open balcony door. The participant can see a visualization of their body’s main skeleton materialize in the virtual room and move in real-time, according to their actual movements. They can interact within the virtual environment with their entire body. Again, the participant has the same three consecutive tasks to execute. They have to walk across the room, towards the open balcony door and get out, on the balcony, touch the railing of the balcony with their two controllers, and stay in this state for a few minutes. Then, they have to catch the object hanging before them, at a significant distance ahead of the balcony railing. When they do so, they can turn back, inside the room, and the task is considered completed. Last, the participant removes the Headset and returns to the real world. A demonstration video is available here: youtube.com/watch?v=9Jqy6GGwg4o.

## 8. Measures

The system used in each experiment was evaluated by the five factors derived by the Presence Questionnaire that was introduced by Witmer and Singer in 1998 [40]. The questionnaire has been revised so as to not include questions irrelevant to the study; in particular, questions about sound feedback have been removed. Each of the questions are relevant to one of the five factors that reflect the level of presence: Control, Sensory, Involvement, Realism, and Distraction factors. More information about the included questions and their significance to the study can be found in the Appendix A. Following, are the five Factors we used to determine the presence each system offers, according to the Witmer–Singer Questionnaire:The Sensory Factor depends on sensory modality, environmental richness, multimodal presentation, consistency of multimodal information, degree of movement perception, and active search. Those criteria relate mostly to the virtual environment itself, and, since the virtual environment used in both system simulations is the same, we do not expect this factor to differentiate the systems.The Realism Factor depends on the realism of the scenes in the virtual environment, the consistency of information with the between the physical and the virtual world, as well as the meaningfulness of the experience for the user and the disorientation after they return to the physical world, at the end of the simulation.The Control Factor reflects the degree of control the participant felt they had during the simulation with each system, depending on immediacy of control, anticipation, mode of control, and modifiability of the physical environment.The Involvement and Distraction Factors are the ones of highest priority for the differentiation between the two systems, and completely aligned with the focus of the study. More specifically, involvement depends on the degree the user feels they interact naturally within the virtual environment, as well as feeling part of it. The degree of distraction depends on isolation of the user from their actual, physical environment, the user’s selective attention toward the simulation and not their physical world, as well as interface awareness, which depends on the interface devices.

## 9. Results

The same Questionnaire was used for evaluation of both the VR-HC and VR-MC system. A questionnaire was handed out to each participant right after they completed the session with the respective system. In this section, we present quantitative and qualitative results derived from the data collected from the Witmer–Singer Questionnaire of each of the VR-HC and VR-MC systems. For each participant, we have derived the value of the aforementioned five factors, by calculating the mean of specific questions for each factor. The questions relevant to each factor, as suggested by Witmer–Singer, go as follows:Control Factor: the mean of the questions 1, 2, 3, 5, 7, 8, 9, 13, 15, 16, and 17.Sensory Factor: the mean of the questions 4, 6, 9, 10, 11, 12, and 21.Involvement Factor: the mean of the questions 4, 10, and 14.Realism Factor: the mean of the questions 7 and 9.Distraction Factor: the mean of the questions 18, 19, 20, 21.

The way in which the values of these factors are interpreted is as follows: the values of each factor range between 0 and 7, using the same seven-point scale format suggested by Witmer and Singer. If the value of a factor is high, then the system is considered to provide a significant sense of presence, regarding this specific factor. Respectively, if the value of a factor is low, then the system is considered to provide a low feeling of presence, based on that factor. 

First, we calculated the Mean and Standard Deviation values for each Factor of each participant (Table 1). Then a paired samples t-test was conducted between the VR-HC and VR-MC systems for each of the factors. The significance level (alpha value) is a = 0.05 and the degrees of freedom are df = 19. So, from the Table of Student t-Distribution, the critical value is cv = 2.0930. Running the t-test, the t-value for each Factor (Control, Sensory, Involvement, Realism, Distraction) is t(20) = 12.354, t(20) = 9.768, t(20) = 18.579, t(20) = 6.850, t(20) = 14.620, p = 0.01 respectively. So, the obtained t-values exceed the critical value by far. That means that all five t-tests are significant and the two systems are statistically significantly different regarding every one of the factors. Especially the Involvement (t(20) = 18.579), Distraction (t(20) = 14.620) and Control (t(20) = 12.354) factors are the ones which highlight the superiority of the VR-MC to the VR-HC system, regarding presence. While the Sensory (t(20) = 9.768) and Realism (t(20) = 6.850) factors, which present the lowest statistical difference between the two systems, are highly related to the quality of the virtual environment, it was not the matter under study. Upon investigating the questions from which the values of the factors are derived, we noticed that in questions 6, 9, 11 and 12 participants gave similar answers (Table 2). That was expected, since questions regarding the Realism and Sensory Factors regard the quality of the virtual environment which was equal in both systems. However, even in those two factors, a significant statistical difference can be observed, because of questions 4, 10, and 21 at the Sensory factor and question 7 at the Realism factor. Those questions regard movement and interaction within the virtual environment, as well as similarities with the real world.

By calculating the mean of each factor for each participant, according to Table 3 and Figure 11, we can observe from an external viewpoint the difference between the VR-MC and VR-HC System. The factors of VR-MC received higher values and, consequently, the VR-MC system generates a significant sense of presence. Observing the results of our statistical analysis, as well as the significant statistical difference between the means of each of the five factors taken into consideration, we can conclude that the proposed system indeed accentuates the feeling of presence of the users.

In this experiment, we examined the impact of presence and interaction during a VRET simulation for acrophobia by comparing a standard VR hand-controller interactive system with a VR full-body motion-capture interactive system. Cross-examining both the information gathered during the relevant literature review and the findings from the study, it can be concluded that the use of the Body and Hand Motion Recognition Cameras in the VR-MC system overall is preferred by users, in comparison to the VR-HC system. By examining the significant statistical differences of the results, especially for the Involvement and Distraction factors, it is safe to conclude that the fact that the motion tracking in the VR-MC system is performed remotely, improves the quality of the simulation, by making the user feel more present and immersed to the virtual environment. Therefore, VR systems can include movement, as well as full body presence without requiring any additional effort on the user’s end, in contrast with current systems, which make movement tracking the user’s responsibility, by adding wearable sensors. In conclusion, the addition of motion recognition cameras was proven of vital importance for the user’s feeling of presence.

## 10. Limitations 

Participants: The sample of the study consisted of only young people (ages 18–30, avg. age 24) which could possibly affect the results since young people are usually more familiar with VR technologies and applications. This means that they may become accustomed to the virtual environment and the experience of moving within it, which of course affects their responses to the questionnaire. In addition, the sample was relatively small, consisting only of 20 people, meaning that results could differ at larger scales.

Procedure: The experiment procedure was short in duration and included only two simple tasks for each participant: moving within the virtual environment and interacting with an object within it. Adding more tasks, as well as including more of the participants’ senses in the tasks could increase their feeling of presence within the systems and help them develop a more holistic opinion about each one of them. In addition, by giving them the chance to use each system in more than one sessions could help them learn how to navigate within each system better. However, the point of this study is, among other things, to highlight the importance of ease when it comes to motion control on behalf of the user, even from their first interaction with the system, since presence may be diminished until the motion controllers are well learnt [48].

System Limitations: Indeed, as with every technological device used for interaction in VR systems, there is no guarantee that all the users will find it equally convenient and immersive. This can be due to their personal experience with technology, which can help them get accustomed to each system more easily or not, as well as the device’s limitations, which are presented below.

The Oculus Hand Controllers were designed to simulate the virtual hands in a very realistic manner, by tracking the position of the controllers, as they move along with the user’s hand movements [49]. Even though the user cannot see their hands in the virtual room, they can see the Hand Controllers moving in real time in it, according to their physical movements. Oculus also provides the option to show actual hands instead of the hand controllers. However, it is important to point out that the hand controllers do not really recognize the hands, even though hands are visible in the virtual environment. In reality, the hand visualization is in accordance with the buttons of the Hand Controllers, which are touch-sensitive. Once the user touches a button in the virtual room, it seems that their virtual finger is moving, but if they move their fingers without touching the buttons, no finger movement appears in the virtual room.

As mentioned in the System Description, the Astra S Model has a range of 0.6–8.0m. So, the Camera Sensor can only recognize the body and limbs of the users, while smaller movements of the hands and fingers are not distinguishable; the hands are visualized as spheres. At the same time, the position tracking accuracy of each point is reported by the camera’s Confidence Factor; if it is above 80%, then it can be considered accurate. If the user stays within 1–3 m from the camera, the tracking is accurate more than 80% of the time. That does not mean that 20% of the time the virtual body is completely out-of-sync or lost in the virtual environment; it means that the virtual movements are not actually not 100% synchronized for a few milliseconds with the user’s actual movements. The difference in position accuracy could be a few millimeters above or below the actual position. The cases of measurements with low confidence factor that can be noticed by the user occur when the user is “hiding” their limbs from the camera and the camera cannot track their joints (i.e., if they place hand behind their back). In other words, the camera works as a mirror; it reflects whatever it sees on a first layer. Of course, the tasks assigned to the participants in this study do not require such movements; all actions that take place can be recognized by the camera. A solution to this issue could be to add another BMRC to the motion tracking system, so that movements that take place away from the first one can still be tracked.

The Leap Motion HMRC allows us to include hands and fingers in the user’s virtual skeleton. Attaching the Leap Motion HMRC to the front of the VR headset is suggested by the manufacturer of the controller, Leap Motion, and a special case for the controller that can be stuck to the front of the VR headset is provided. That particular placement of the controller allows the user to view their virtual hands when they are placed in front of them, and the user is looking toward them. That manner of tracking covers most cases of movement when the hands of the user could be in their visual field, even in real life. A scenario in which their hand could not be visible in the virtual environment could occur if the user looked at a different direction than the one their hand is and moves. In real life, thanks to peripheral vision, the user could see their hand work, but in the virtual environment that will not be possible, since the hand will not be in the tracking field of the Leap Motion HMRC.

Additionally, it is important to point out that tracking the movements of a user accurately in order for them to interact with objects in the virtual environment is not enough to increase presence. Touch is a significant sense of the human body and creating a user interface that could apply touch is crucial to improve virtual reality presence. For our current research, Leap Motion HMRC was the middle step, in order to find the value of life-like virtual interaction but does not provide a life-like touch sense of interaction. Current technologies such as Gloves, Suits, and Hand Controllers provide mechanical feedback which is relatively uncomfortable and falls under the original assertion of this study. Although forthcoming technologies such as ultrahaptic feedback [50] could provide the seance of touch without out wearable equipment.

## 11. Discussion

After examining the results of our study, we concluded that transferring the means of interaction away from the user and making it the system’s responsibility to provide interaction resulted in increased presence. Additional and wearable equipment adds an extra burden on the user, affecting not only the interaction but also the user’s behavior during the simulation, and, consequently the way they confront the stimuli. Overall, if properly tracked and presented, the hands are the most natural user interface a system can use to allow the user to interact within it, while the presence of the whole body inside the virtual world gives the user a sense of their virtual existence. 

Furthermore, the purpose of our system is to be used by psychologists and psychiatrists as an additional tool to help their patients learn useful skills in order to confront their source of anxiety; in this case, acrophobia. A person that already finds themselves in an anxious environment should be able to at least interact within it in a natural manner and not face the extra anxiety of using the simulation system correctly. This is another reason why it is preferable to include non-wearable tracking methods in such systems.

By analyzing the results of our study, we observed that even though the feeling of presence provided by the VR-MC system was considerably better than that of the VR-HC system, there are still several issues to solve and obstacles to overcome until we come up with a completely immersive VR system for the treatment of phobias and mental illnesses in general. The fact that we are using a single BMRC means that in case the user would have to engage in more complex tasks that require moving in a wider virtual environment, the range of the camera would not be enough. Hence, we propose for further study the addition of more than one BMRCs to provide motion tracking from different angles and allow the user to move completely freely in the enclosing space. Last but not least, reviews [35] highlight that using wireless VR headsets could ensure more comfortable navigation for users in the physical environment during the simulations, so, we also suggest that wireless or mobile VR headsets shall be used in future versions of such systems to examine this matter.

Overall, considering the technical issues of Standard ET such as affordability, the time needed to applicate it, as well as the fact that it is impossible for clinicians to acquire every phobic stimulus possible to apply it, VRET seems to be at least as helpful as Standard ET. Acquisition costs of VR systems have dropped significantly, making it possible for VRET to be applied in a larger scale in clinicians’ offices. By continuing researching the potential of systems like the one under study, we could come up with better combination of sensors to actually enhance the treatment methods and help more patients battling mental illness in the long run.

## Figures and Tables

**Figure 1 sensors-20-01244-f001:**
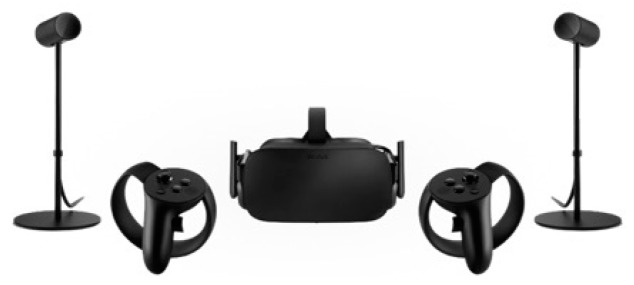
Peripheral hardware of the Virtual Reality Hand Controllers (VR-HC) system.

**Figure 2 sensors-20-01244-f002:**
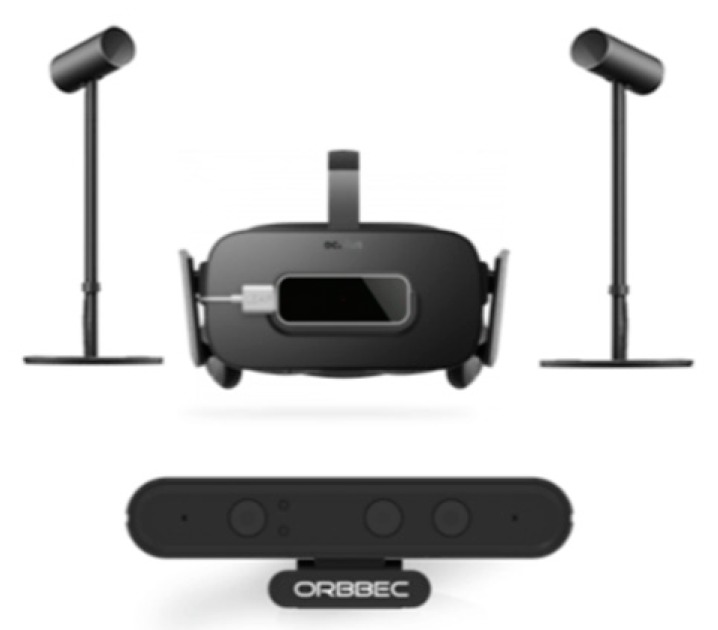
Peripheral hardware of the Virtual Reality-Motion Camera (VR-MC) system.

**Figure 3 sensors-20-01244-f003:**
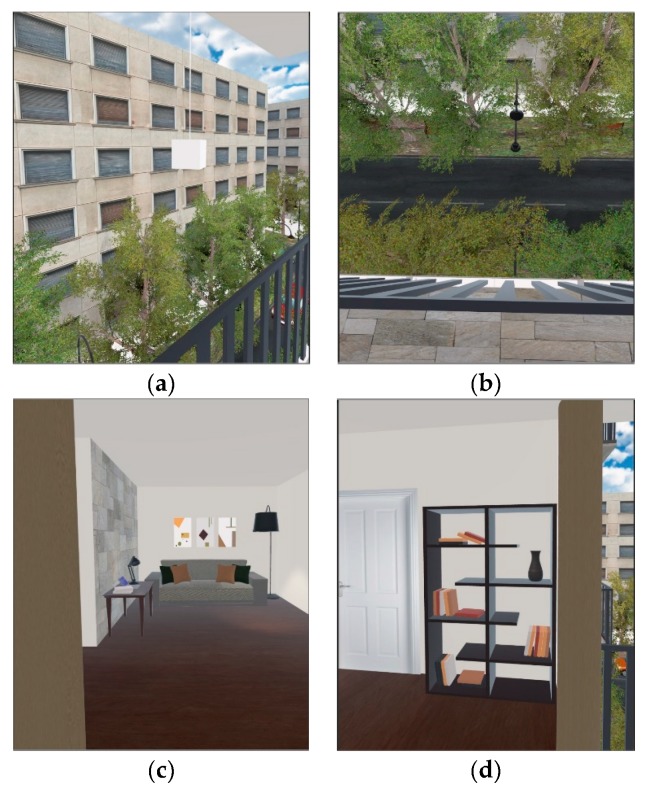
(**a**–**d**) The Virtual Reality environment used in both systems.

**Figure 4 sensors-20-01244-f004:**
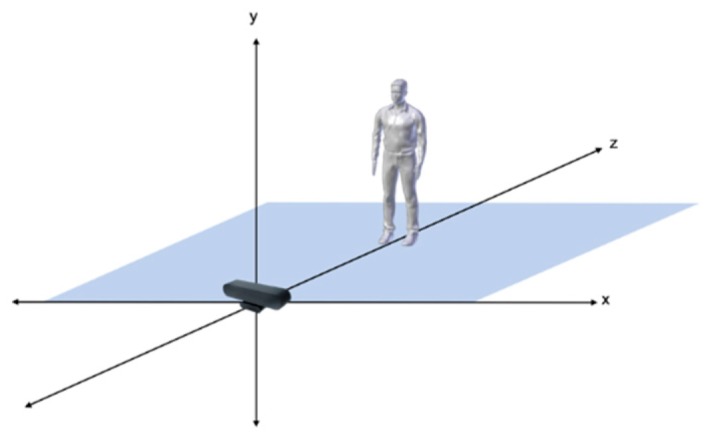
The axis topology.

**Figure 5 sensors-20-01244-f005:**
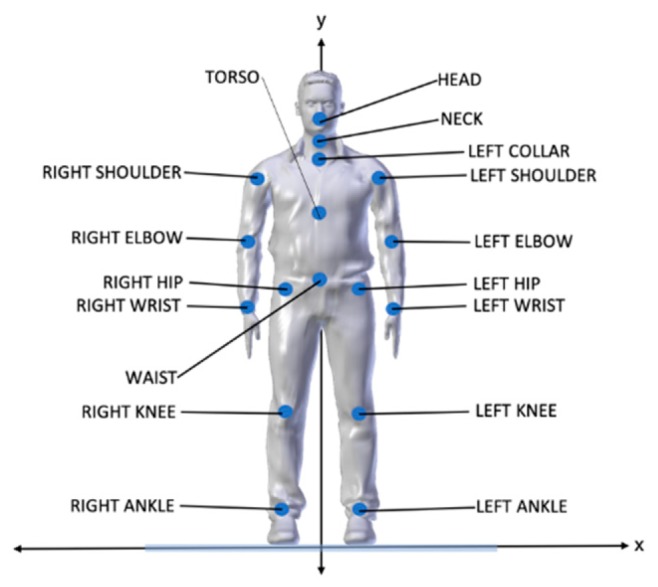
The 17 recognizable joints of the Astra S.

**Figure 6 sensors-20-01244-f006:**
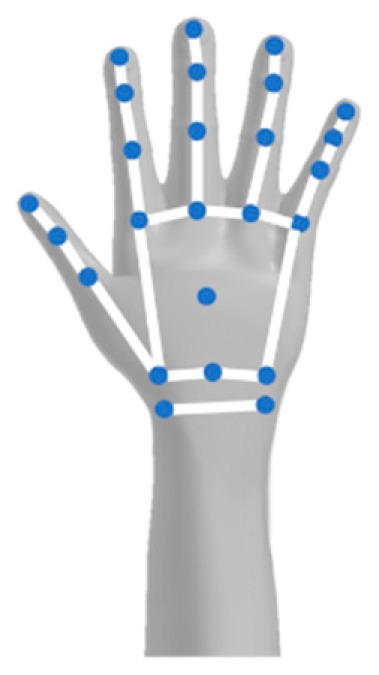
The 25 recognizable joints of the Leap Motion.

**Figure 7 sensors-20-01244-f007:**
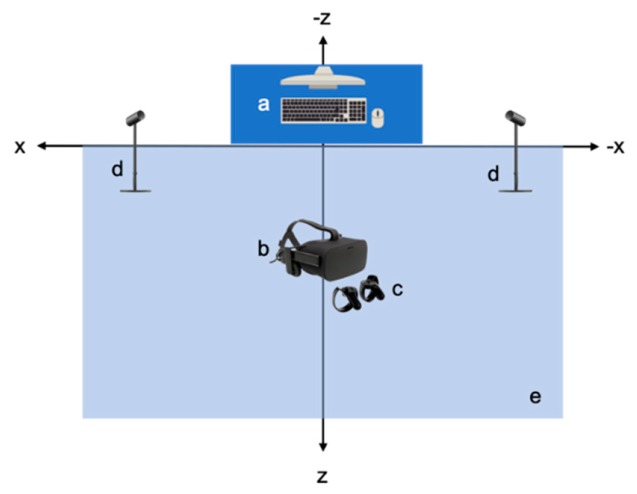
Setup of the VR-HC System: (**a**) Desktop Computer, (**b**) Oculus Rift VR Headset, (**c**) Oculus Rift Hand Controllers, (**d**) Oculus Rift Sensor, (**e**) VR-HC User Action Area.

**Figure 8 sensors-20-01244-f008:**
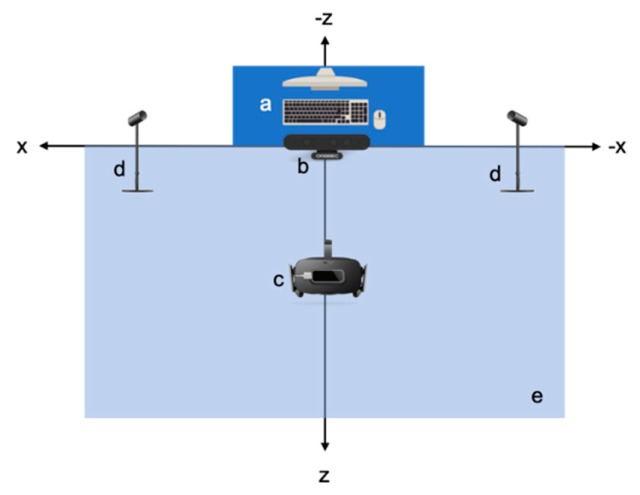
Setup of the VR-MC Tracking System: (**a**) Desktop Computer, (**b**) Astra S Body Motion Recognition Camera (BMRC), (**c**) Oculus Rift VR Headset with attached Leap Motion Controller Hand Motion Recognition Camera (HMRC), (**d**) Oculus Rift Sensor, (**e**) VR-MC User Action Area.

**Figure 9 sensors-20-01244-f009:**
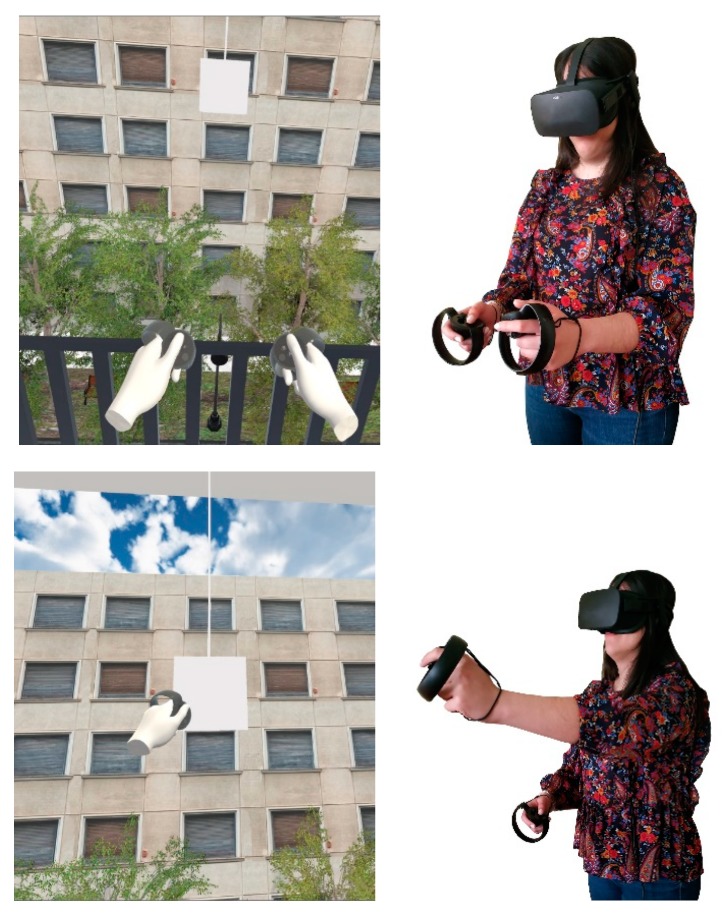
Demonstration of the VR-HC simulation during the procedure.

**Figure 10 sensors-20-01244-f010:**
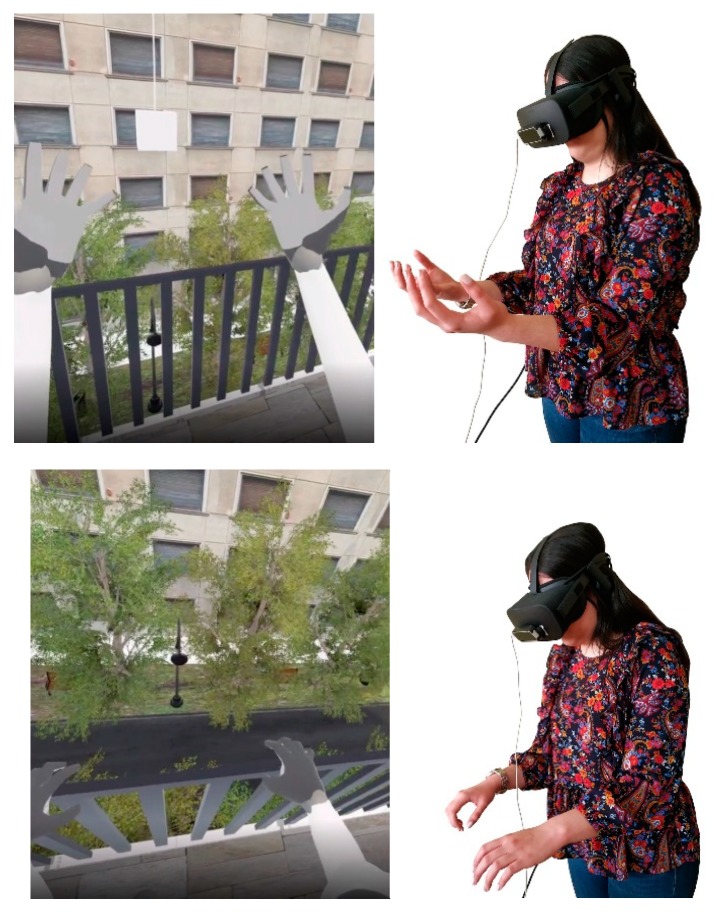
Demonstration of the VR-MC simulation during the procedure.

**Figure 11 sensors-20-01244-f011:**
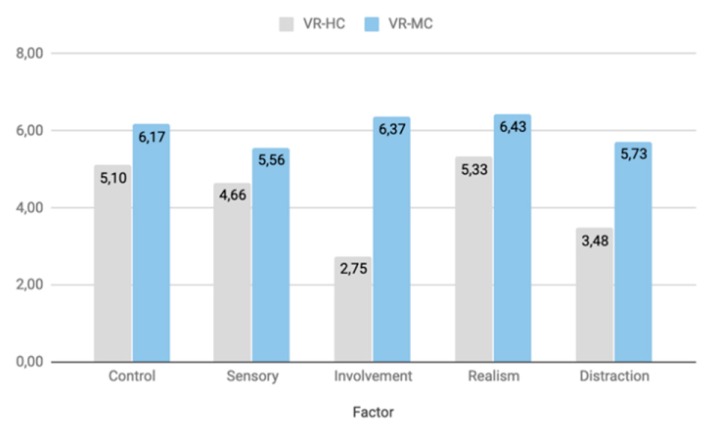
Comparison of the mean of the Control, Sensory, Involvement, Realism, and Distraction Factors between the VR-HC and VR-MC systems.

**Table 1 sensors-20-01244-t001:** The Mean and Standard Deviation of the Control, Sensory, Involvement, Realism, and Distraction Factors for each participant for the VR-HC and VR-MC systems.

*Participants Data*	Control Factor				Sensory Factor				Involvement Factor				Realism Factor				Distraction Factor			
	VR-HC		VR-MC		VR-HC		VR-MC		VR-HC		VR-MC		VR-HC		VR-MC		VR-HC		VR-MC	
	Mean	SD	Mean	SD	Mean	SD	Mean	SD	Mean	SD	Mean	SD	Mean	SD	Mean	SD	Mean	SD	Mean	SD
*part. 1*	4.73	1.90	6.45	0.82	4.50	2.71	5.71	4.37	1.50	0.58	6.00	1.00	5.00	1.41	7.00	0.00	2.50	3.00	6.25	0.96
*part. 2*	4.91	1.92	4.37	0.70	4.50	2.41	5.29	1.25	1.50	0.58	5.67	0.58	5.00	1.41	6.00	1.41	3.00	2.00	5.75	1.89
*part. 3*	4.91	1.51	6.18	0.87	4.83	1.98	5.29	1.50	3.50	0.58	6.00	1.00	5.50	0.71	6.50	0.71	3.25	2.63	5.50	2.38
*part. 4*	5.27	1.42	5.73	0.79	4.83	2.56	6.00	1.15	2.50	1.00	6.33	0.58	5.00	1.41	6.00	0.00	3.00	2.16	6.00	1.41
*part. 5*	5.45	4.35	6.55	0.69	5.00	2.23	5.71	1.60	2.00	0.00	6.67	0.58	6.00	1.41	6.50	0.71	4.25	0.96	5.50	1.73
*part. 6*	4.73	1.42	6.18	0.40	4.33	2.38	5.43	1.51	2.50	1.53	6.00	1.00	4.50	4.38	6.00	0.00	2.75	1.71	5.75	1.89
*part. 7*	5.00	1.55	6.73	0.65	4.50	2.58	5.71	1.98	3.50	1.53	6.33	0.58	6.00	1.41	7.00	0.00	3.00	2.00	5.75	2.50
*part. 8*	5.00	1.26	6.00	0.77	4.17	2.45	5.86	1.46	1.50	1.00	6.33	0.58	5.50	0.71	6.50	0.71	4.25	1.26	6.00	0.00
*part. 9*	4.37	1.76	6.27	0.90	5.00	1.98	5.57	1.62	3.00	0.00	6.67	0.58	5.50	4.38	6.00	1.41	3.75	2.22	5.75	1.89
*part. 10*	4.91	1.92	6.00	0.63	4.33	2.19	5.71	0.95	3.00	1.53	6.00	1.00	5.00	1.41	6.00	0.00	4.00	2.16	5.75	1.50
*part. 11*	5.36	1.50	6.27	0.79	4.67	1.80	5.29	1.38	3.50	1.00	6.00	1.00	5.00	2.83	6.50	0.71	3.50	1.73	6.00	1.41
*part. 12*	5.18	1.60	6.00	1.00	4.33	1.95	5.57	1.90	3.50	1.53	6.67	0.58	4.50	4.38	6.50	0.71	4.25	1.26	5.00	2.16
*part. 13*	5.91	1.38	6.27	0.79	5.00	1.99	4.86	4.38	3.50	1.53	6.00	1.00	5.50	4.38	7.00	0.00	4.25	2.22	5.75	2.50
*part. 14*	5.18	1.54	6.36	0.92	4.83	1.72	5.43	2.44	3.00	0.58	6.33	1.15	6.00	1.41	6.50	0.71	3.50	4.37	5.25	1.50
*part. 15*	5.18	0.87	6.18	0.87	5.00	1.70	6.00	1.15	4.00	0.58	6.67	0.58	4.50	0.71	6.50	0.71	3.75	1.71	5.75	1.50
*part. 16*	4.91	1.81	4.37	0.83	4.67	1.86	5.29	1.70	3.00	1.15	7.00	0.00	4.50	4.38	6.00	0.00	3.75	2.50	6.00	2.00
*part. 17*	4.64	4.38	4.37	0.94	4.33	2.45	5.71	1.98	1.50	0.58	6.67	0.58	4.50	0.71	7.00	0.00	3.00	2.16	5.75	2.50
*part. 18*	5.45	1.69	5.82	1.33	5.00	2.15	5.71	2.21	3.50	1.53	6.67	0.58	6.50	0.71	6.50	0.71	3.50	1.91	6.00	2.00
*part. 19*	4.37	1.51	5.82	0.98	4.67	2.15	5.43	1.40	3.00	0.00	6.33	0.58	6.50	0.71	6.50	0.71	3.25	2.63	5.50	1.73
*part. 20*	5.18	1.40	6.27	0.79	4.67	2.57	5.71	2.21	2.00	1.15	7.00	0.00	6.00	1.41	6.00	1.41	3.00	1.83	5.50	2.38

**Table 2 sensors-20-01244-t002:** Comparison of the mean of Questions 6, 9, 11, 12, between the VR-HC and VR-MC systems.

Question	VR-HC	VR-MC
6	6,45	6,4
9	6,3	6,35
11	6,25	6,25
12	3,75	3,8

**Table 3 sensors-20-01244-t003:** Comparison of the mean of the Control, Sensory, Involvement, Realism, and Distraction Factors between the VR-HC and VR-MC systems.

Factor	VR-HC	VR-MC
Control	5,10	6,17
Sensory	4,66	5,56
Involvement	2,75	6,37
Realism	5,33	6,43
Distraction	3,48	5,73

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
