# Peer review of "Comparison between Full Body Motion Recognition Camera Interaction and Hand Controllers Interaction used in Virtual Reality Exposure Therapy for Acrophobia"

_sensors, 2020, doi:10.3390/s20051244_

Round 1

Reviewer 1 Report

This article proposes a comparison of two immersive virtual reality devices for patients with acrophobia, one is the standard commercially available Oculus Rift S + Touch VR, and the other is the combination of Orbbec Astra S depth camera for full-body motion recognition, plus Leap Motion gesture recognition VR system.

The results show that VR-MC has a better sense of presence, while there is no additional wearable device that will increase the user's burden, and it can adopt natural interactive operations in the virtual reality. This paper shows that VRET using VR-MC can be used as a supplement to Standard ET.

VR-MC integrates Leap motion and BMRC. The use of such hardware combinations in virtual reality is not new. Not very specific specifications or requirements, so the description of the desktop computer is redundant. (line 149 - line 152) Only 20 experimenters, the sample size is very low, especially when reference 30. There are 180 samples (line 553), 31. There are 100 participants (line 555), It is necessary to explain the representative of the 20 samples of this study. No need to describe statistical terms. (line 318 -323) Please pay attention to the correctness of the word, for example, "hole" body is guessed as "whole" body (line 446). The author believes that VR-MC's non-wearable tracking methods improve presence and immersion. It can reduce the user's anxiety and burden, but it is also suggested that in order to simulate the tactile sensation, an additional tactile feedback device (line 437-438) can be added in the future, which is obviously contradictory.

Author Response

Dear Reviewer, 

Thank you for your feedback and your comments. 

Concerning your point: “The use of such hardware combinations in virtual reality is not new. Not very specific specifications or requirements, so the description of the desktop computer is redundant. (line 149 - line 152)”. You are right, in our study was not clear and for that reason we added this phrase (lines 136-140):  

In more technologically oriented studies, the use of various tracking tools that can help improve the human experience in the virtual augmented or mixed reality world have been thoroughly studied [42]–[46]. Nevertheless, there is still much room for research on how to apply these technologies in the fields related to mental disorders and, more specifically, what tool can be used in diverse types of treatment.

Concerning your point: “Only 20 experimenters, the sample size is very low, especially when reference 30. There are 180 samples (line 553), 31. There are 100 participants (line 555), It is necessary to explain the representative of the 20 samples of this study.” Yes you are absolutely right the sample size is relatively small and we emphasized this in the limitation section (lines 37-382) however even with this sample size the results are discrete and hope in the future to acquire more resources to execute the trials with improvements on the system (limited resources was our core problem and could not do more).    

Concerning your point: “No need to describe statistical terms. (line 318 -323) ”. We deleted those lines. 

Concerning your point: “Please pay attention to the correctness of the word, for example, "hole" body is guessed as "whole" body (line 446).” Sorry for that. Done. 

Concerning your point: “The author believes that VR-MC's non-wearable tracking methods improve presence and immersion. It can reduce the user's anxiety and burden, but it is also suggested that in order to simulate the tactile sensation, an additional tactile feedback device (line 437-438) can be added in the future, which is obviously contradictory.” You are right, was not clear and for that reason we added this phrase (lines 435–438):  

Current technologies such as Gloves, Suits and Hand Controllers provide mechanical feedback which is relatively uncomfortable and falls under the original assertion of this study. Although forthcoming technologies such as ultrahaptic feedback [50] could provide the seance of touch without out wearable equipment. 

Reviewer 2 Report

This paper addresses an interesting topic, the use of virtual reality to implement Exposure Therapy. VR allows to simulate phobic environments without restrictions about the feared stimuli and cost.

Indeed, users can interact safely and have access to a large range of situations.

The paper is well written and well structured. The materials (figures and tables) are adequate. It is necessary to correct references 1 and 28. Figure 3 will be better placed in page 5.

I disagree with the definition of immersion presented line 123. Immersion is related to « replacement » of perception from real world. A more formal and pertinent definition can be found in reference from Witmer and Singer:  « Immersion is a psychological state characterized by perceiving oneself to be enveloped by, included in, and interacting with an environment that provides a continuous stream of stimuli and experiences ». Technology is only one factor affecting immersion.

The paper details properly the user study and its limitations. In particular, spatial issues that could be improved with wireless headsets and more than one camera for motion tracking. I think it would be interesting to take into account the body representation which is different for the two cases VR-HC and VR-MC.

Author Response

Dear Reviewer, 

Thank you for your feedback and your comments. 

Concerning your point: “The materials (figures and tables) are adequate. It is necessary to correct references 1 and 28.” Done

Concerning your point: “”

Concerning your point: “Figure 3 will be better placed in page 5”. Done

Concerning your point: “I disagree with the definition of immersion presented line 123. Immersion is related to….” Yes you are right, we wrote the immersion “definition” from a tech perspective which ended to be fault and we changed it as Witmer and Singer wrote it.

Concerning your point: “spatial issues that could be improved with wireless headsets and more than one camera for motion tracking….” Yes you are right the prof Koutsouris sed same ideas but the study score and size would be increased a lot if we wanted to compare those factors also .

Moreover we changed and added the the lanes: (lines 136-140) / (lines 435–438)